# Research on the characteristics and influencing factors of the Beijing-Tianjin-Hebei urban network structure from the perspective of listed manufacturing enterprises

Wengang Wang[1,2,3], Chuning Miao[1]*, Haihang Yu[1], Can Li[1,2]

1 School of Geographical Sciences, Hebei Normal University, Shijiazhuang, Hebei Province, China, 2 Hebei Key Laboratory of Environmental Change and Ecological Construction, Hebei Normal University, Shijiazhuang, Hebei Province, China, 3 Hebei Technology Innovation Center for Remote Sensing Identification of Environmental Change, Hebei Normal University, Shijiazhuang, Hebei Province, China

* 18731065581@163.com

**Data Availability Statement:** All relevant data are within the paper and its Supporting information files.

## Abstract

The production connection is a crucial component of the Inter City Association. An urban network based on the enterprise perspective better reflects the structural characteristics of regional cities. Based on data gathered from the headquarters and branches of the listed manufacturing enterprises in 2020, this paper analyzes the county-level administrative units in the Beijing-Tianjin-Hebei region of China. Using the subordinate connection model and the social network analysis method, this paper examines the spatial structural characteristics and factors of urban networks in the Beijing-Tianjin-Hebei region. The results suggest that resource allocation in the Beijing-Tianjin-Hebei region is unbalanced, with a significant difference in urban radiation and agglomeration capacity. As the administrative centers, Beijing, Tianjin, and Shijiazhuang show a strong ability to allocate resources within the network. The overall network density in the region was shown to be relatively low, with the main links being of low or medium level. The urban network is defined by the network connection led by Beijing and Tianjin with Beijing-Tianjin-Tangshan and Beijing-Baoding-Shijiazhuang as the main axis. These cities exhibit a "dense southeast and sparse northwest" pattern. From a hierarchical perspective, high-level network connections are based mainly on spatial proximity. Analyzing the agglomerative subgroups, the study found that the inner and outer connections of the core subgroups were relatively high. Semi-marginal subgroups generally accepted the radiation of core subgroups, while marginal subgroups had little connection with other subgroups in the region. The results of the QAP analysis show that the administrative relationship, spatial distance, city size difference, economic development level difference, industrial structure difference, and labor cost difference have a significant influence on the urban network of the Beijing-Tianjin-Hebei region.

**Funding:** Science and Technology Research Fund of Hebei Normal University: Structural Evolution and Regional Collaborative Development of Beijing-Tianjin-Hebei Urban Economic Network (Project No. L2020Z07). The funders had no role in study design, data collection and analysis, decision to publish, or preparation of the manuscript.

**Competing interests:** The authors have declared that no competing interests exist.

# Introduction

Since the 21$^{st}$ century, China's rapid urbanization has triggered a multi-scale urban spatial reconstruction. Developments in transportation and communications mean that mobility between cities is also constantly advancing and intertwining to form a complex urban flow network. Urban networks strengthen the interdependence between cities and help identify important nodes in network research [1, 2].

The urban network research is based on three notions, namely Stephen Hymer's prediction of the global city grade [3], Friedmann's "world city hypothesis" [4] and Saskia Sassen's concept of a global city [5]. However, these notions are based on data on city attributes, meaning that little or no attention is paid to the relationship between cities [6]. Castell's theory of "mobile space" emphasizes the role of the city as a node and hub within mobile space, laying the theoretical foundation for the later study of the urban network [7]. The relational data represented by stream data shifted the situation so that attribute data no longer had an absolute precedence in urban network research. The relational data can show the multidimensional social, economic, and cultural connections between cities. Moreover, this type of data can also be used to mine urban network structure based on traffic flow data from aviation, railways, and highways [8–10], information flow data from mobile phone signaling, Internet traffic, and Weibo [11–13], and enterprise data [14]. However, traffic flow and information flow data are not able to thoroughly examine the function delimitation, spatial governance, and construction of the city network [15]. In contrast, as an important carrier of capital, information, and talent, the inter-regional layout of enterprises has become key force shaping the urban network.

In recent years, scholars have carried out multidimensional research on the urban network organization. Scholars have used different types of enterprise data to study urban networks. Holl and Mariotti focused on the relationship between location, accessibility, and the urban structure of logistics enterprises [16]. On the other hand, Rozenblat et al. studied the regional development of cities based on the economic links between transnational corporations and their industrial sectors [17]. Furthermore, Yin Jun, Zhao Jinli et al. used financial services data to examine China's urban network [18, 19]. Jiang Xiaorong, Sheng Kerong, Zhong Yexi et al. gathered data from listed enterprises to explore the structure and centrality of the urban network in China [20–22]. In addition to the research on the spatial connection pattern, network structure, and factors affecting the urban network, research on network resilience [23, 24] and network evolution [25, 26] has also been conducted. There are two types of enterprise-based urban network research methods. The first is the "Globalization and World City network research group" (GaWC) represented by Taylor. It predominantly constructs chain network models based on advanced producer service enterprises [27]. The other is to take Alderson and other scholars as representatives in order to construct the city network according to the affiliation of headquarters and their subsidiaries and giving importance to the affiliation between headquarters and branches [28]. The first method focuses on advanced producer services and is sensitive to the relative importance of enterprises in cities. Conversely, the latter focuses on multinational companies in all industrial sectors and pays more attention to hierarchical differences within enterprises [29]. Within the chain network model, all enterprise branches are connected, even though this is not the case in reality. Rozenblat et al. argue that the relationship within the enterprise is established primarily between headquarters and branches, leaving fewer connections between the branches themselves [17]. Since the chain network model cannot calculate the city centrality and redundant connection, it is unable to explore the urban network in detail [30]. However, the membership relation model is able to realize the above operation. Therefore, this paper uses the latter model when building the city network.

Scale is an important characteristic of urban networks. As the spatial scale decreases, the complexity of the urban network structure increases. At present, most research on urban networks is conducted at the global or national level, paying little attention to the regional city networks of a country. In addition, most existing studies are based on provincial or municipal research units. There are few studies conducted at the county level [31, 32]. Urban network research at the regional scale in China has focused on the Beijing-Tianjin-Hebei region, the Yangtze River Delta, and the Pearl River Delta [33]. As one of the key national development areas, research on the Beijing-Tianjin-Hebei urban network has gradually drawn the attention of scholars. However, most of these studies based on the enterprise perspective are still not detailed enough. The average scale of the listed manufacturing enterprises is large, and they generally have high brand value and market influence. The geographical distribution of this type of enterprise largely reflects the regional development potential and economic development, and the data are easily accessible and reliable. Therefore, this paper selects Beijing, Tianjin and Hebei as its research area, taking the county as its basic research unit. The study uses data on the headquarters-branch relationship of the listed manufacturing enterprises to analyze the spatial structure characteristics of the urban network and explore the relationships established between the cities.

## Survey, data sources and research methods

### Overview of the study area

The Beijing-Tianjin-Hebei region includes Beijing, Tianjin, and Hebei Province. There are a total of 13 cities at the prefecture level or higher (Fig 1). Together, they cover an area of

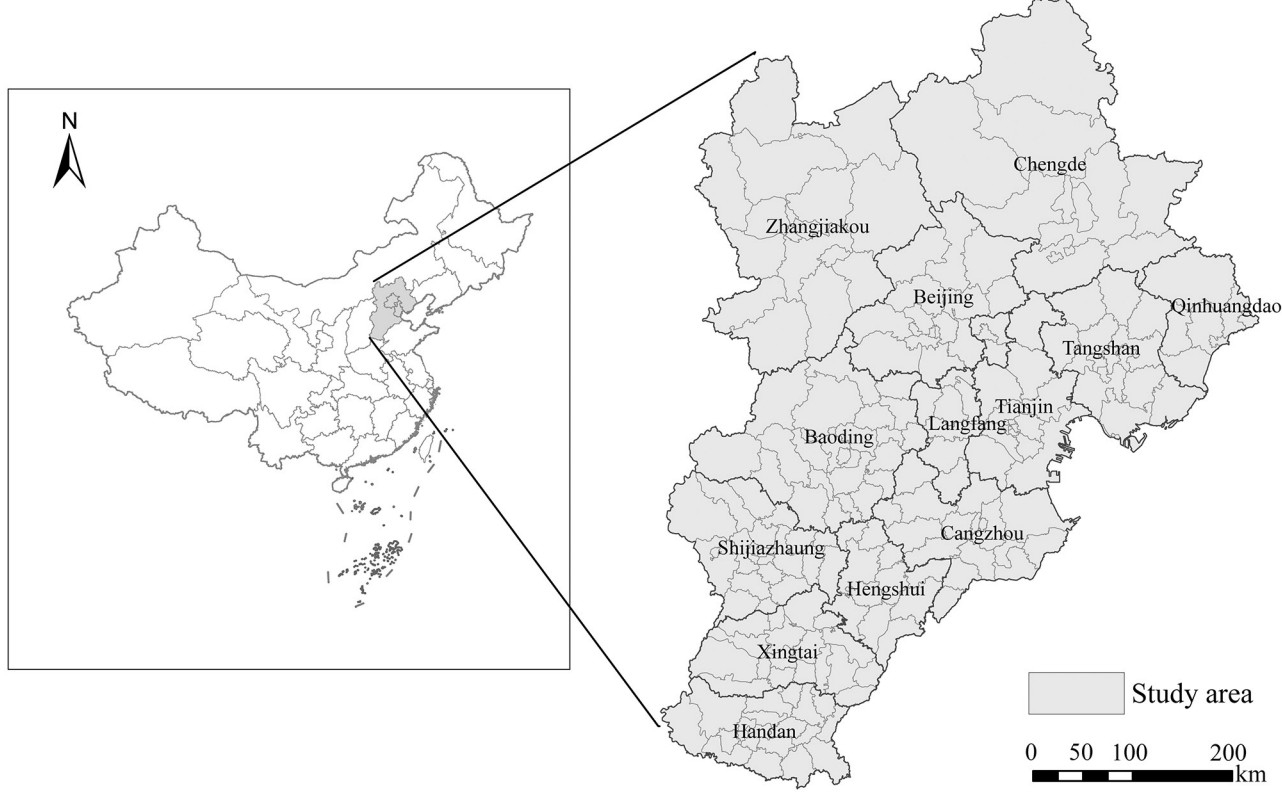

**Fig 1. Map of the Beijing-Tianjin-Hebei region.** (The maps provided in this manuscript are all map images made by myself through software. The images are similar to but not identical to the original images, so they are only used for illustrative purposes).

216,000 km$^2$, accounting for 2.3% of the national land area. This area is high in the northwest and low in the southeast. It belongs to a temperate continental monsoon climate. By the end of 2020, the permanent population of the Beijing-Tianjin-Hebei region was about 110 million, accounting for 7.82% of the entire country. Its GDP was 8.6 trillion yuan, accounting for 8.5% of the country's total GDP. The Beijing-Tianjin-Hebei region is the most dynamic part of northern China. It has industrial strength and many manufacturing enterprises. Therefore, it represents an important manufacturing base in the country [34]. After the State Council issued "Made in China 2025" in May 2015, a coordinated development strategy for the Beijing-Tianjin-Hebei region was put forward in June 2015, which significantly accelerated the development of the manufacturing industry in the region. In 2020, the industrial added value of Beijing is estimated at 421.65 billion yuan, while that of Tianjin is 418.813 billion yuan, and that of Hebei Province is 1154.59 billion yuan. The Beijing-Tianjin-Hebei region is pivotal in promoting China's economic growth. The research on listed manufacturing enterprises within the region will help us better understand the characteristics of its urban production network.

## Data sources

The listed manufacturing enterprises represent excellent enterprises. In order to expand, the listed companies seize market share and set up regional and global branches, thus creating a complex enterprise network. The research data of this paper were extracted from the headquarters of the listed manufacturing enterprises in the Beijing-Tianjin-Hebei region in 2020 through the CSMAR database. Basic information such as name, region, type of industry, address, shareholding relationship, and establishment date of headquarters and branches are obtained through the Tianyancha website, Enterprise investigation website and official enterprise websites. The longitude and latitude of headquarters and branches are obtained through Baidu's map coordinate picking system. Enterprises whose headquarters are not in Beijing, Tianjin, and Hebei are excluded. In addition, enterprises that were cancelled, revoked, closed, or have no branches in the Beijing-Tianjin-Hebei region were also excluded from the study. The branches of the listed manufacturing headquarters in this paper are limited to five types: subsidiary company, branch company, joint venture company, associated company, and other types. Due to the comprehensiveness and authenticity of the data, there is no screening or elimination of equity participation relationship contained in branches. The study area includes 199 district and county administrative units located in 13 prefecture-level cities. Finally, 140 headquarters and 465 branches of the listed manufacturing enterprises fulfilled the requirements of urban network analysis.

## Research methods

**Social network analysis method.** *Network density*. Network density denotes the degree of connection between cities in the network. The higher the network density, the closer the connection between cities will be. The formula for calculating network density is as follows:

$$D_n = L/[n(n-1)] \tag{1}$$

In the formula above, $D_n$ represents the density of the urban network, while $L$ denotes the actual number of connections in the urban network. Next, $n$ represents the number of nodes within the network, i.e., the number of cities. The network density is between 0 and 1. The closer this number is to 1, the higher the network density will be.

*Centrality analysis*. The centrality degree index is predominantly used in this analysis. Degree centrality denotes the number of nodes that are directly connected to one node in the network. It reveals whether the node in question is at the core position within the network. A

higher degree centrality will signal greater rights of the node. An adjacency matrix is constructed based on the relationship between headquarters and branches, while the UCINET software is used to measure the degree centrality of each node, thus showing the status and role of each city in the Beijing-Tianjin-Hebei Urban Network. Since this urban network is directional, the degree of each point may be divided into a point out degree and a point in degree. The point out degree represents the number of other points entering the point, while its size denotes the radiation capacity of the city. Conversely, the point in degree represents the number of relationships directly sent by the node, and its size represents the agglomeration capacity of the city.

*Cohesive subgroup analysis*. An agglomerative subgroup denotes the set of actors with a relatively strong, close, regular, or positive relationship within the network. Inter-city correlation is more likely to occur in the agglomerative subgroup. The research of agglomerated subgroups is able to simplify the network structure, reveal the number of these subgroups within the network and the number of city members contained in each subgroup, explain the development of urban networks, clarify the status and role of city nodes within the network, and distinguish which cities form close relationships [35].

*QAP analysis*. In traditional multiple regression analysis, multiple independent variables must be independent of each other and cannot be highly correlated. Otherwise, multiple collinearity problems occur, affecting the prediction ability of the model. The data in the city network are inherently correlated, meaning that conventional statistical testing methods cannot be used to examine whether there is a correlation between the two types of relations. QAP is a nonparametric method used to study relational data and does not require independent variables. It is able to explain the relationship between cities, thus testing the relationship variables [36]. Therefore, this paper uses the QAP analysis method to explore the factors affecting the spatial structure of the Beijing-Tianjin-Hebei Urban Network.

**Urban connection strength and node strength.** Based on the existing literature, the strength of the factor connection between the headquarters and its subsidiaries, joint ventures, associates, and the other types of affiliated enterprises was assigned [37]. The assignment proportion is 3:2:1:1:0.5. According to this assignment scheme, the city connection degree formed by the headquarters and their branches can be calculated by the following formula:

$$T_{i \rightarrow j} = \sum_{k=1}^{n} (3T_{zk} + 2T_{sk} + T_{hk} + T_{lk} + 0.5T_{qk}) \tag{2}$$

In formula (2), $T_{ij}$ denotes a one-way contact strength between the headquarters located in city $i$ and the branches located in city $j$. Next, $T_{zk}$, $T_{sk}$, $T_{hk}$, $T_{lk}$, and $T_{qk}$ represent the number of subsidiaries, sub-subsidiaries, joint ventures, associates, and other enterprises of the k$^{th}$ listed manufacturing enterprise whose headquarters are situated in city $i$ and branches in city $j$. Moreover, the coefficient denotes the strength of contact between the headquarters and its branches. After calculating the one-way connection strength between two cities, the total connection strength and node strength between the two cities are calculated as follows:

$$R_{ij} = T_{ij} + T_{ji}, G_i = \sum_{j=1}^{n} R_{ij}, (i \neq j), (i = 1, 2, 3, \ldots . n) \tag{3}$$

In formula (3), $T_{ij}$ represents a one-way connection between the enterprise headquarters located in city $i$ and the branches located in city $j$. Next, $T_{ji}$ denotes a one-way connection between the enterprise headquarters situated in city $j$ and the branches situated in city $i$. The symbol $R_{ij}$ represents the sum of interconnections between city $i$ and city $j$, i.e., the connection

strength between the two cities. The sum of the connections between city $i$ and all other cities within the region represents the node strength of the city, $G_i$.

## Spatial distribution characteristics of the listed manufacturing enterprises in the Beijing-Tianjin-Hebei region

### Spatial distribution of manufacturing enterprises

The spatial distribution of the listed manufacturing enterprises in the Beijing-Tianjin-Hebei region is unbalanced. More specifically, headquarters and branches are more concentrated in areas with a higher level of economic development. Zhangjiakou and Chengde represent important ecological barrier cities in the northern part of the Beijing-Tianjin-Hebei region, limiting the development of manufacturing enterprises in this area (Fig 2). The headquarters of the 140 listed manufacturing enterprises are distributed across 54 districts and counties, among which Haidian District, Changping District, Fengtai District and Daxing District in Beijing comprise the largest number of enterprises, accounting for 40% of the sample. These districts are followed by Xiqing District and Binhai New Area in Tianjin, and Chaoyang District in Beijing, which account for 13% of the sample. A total of 465 branches are distributed across 102 districts and counties. Areas with a large number of enterprises include Dongli District, Xiqing District, Binhai New Area and Beichen District in Tianjin, Caofeidian District and Fengnan District in Tangshan, Jingxiu District in Baoding, and Daxing District, Shunyi

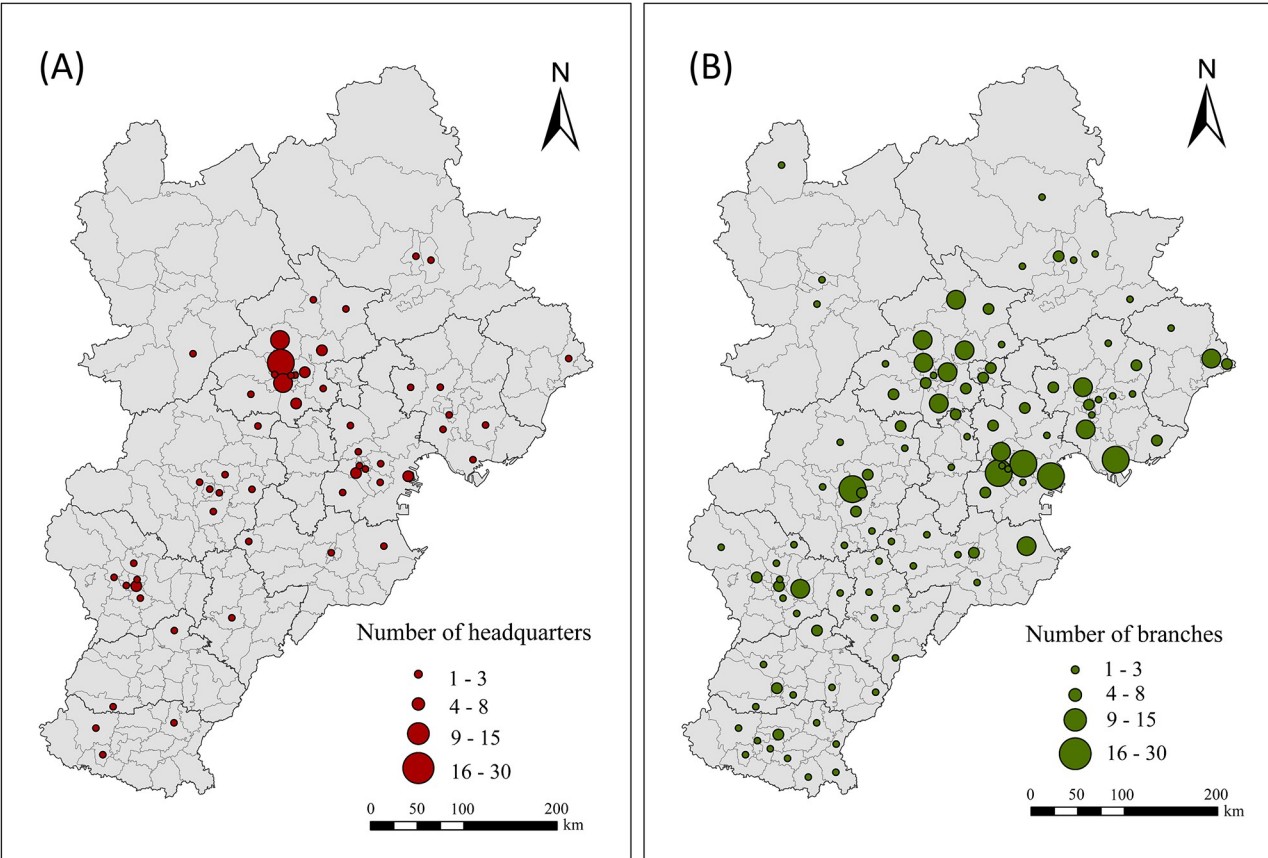

**Fig 2. Spatial distribution of the listed manufacturing enterprises in Beijing, Tianjin and Hebei.** (A) Number of headquarters, and (B) Number of branches.

District, Changping District and Haidian District in Beijing, which together account for 37% of the sample.

## Spatial distribution of manufacturing enterprises by type

The industry types of manufacturing enterprises are classified according to the industry classification guidelines for listed companies issued by the Chinese Securities Regulatory Commission in 2012. There are a total of 27 types of industries based on the headquarters and branches of the listed manufacturing enterprises in the Beijing-Tianjin- Hebei region. According to Xu Weixiang [38], manufacturing enterprises are divided into four categories: technology-intensive, labor-intensive, capital intensive, and resource-intensive (Table 1).

The number of technology-intensive enterprises is the highest in the Beijing-Tianjin-Hebei region (Fig 3). Beijing is the center of pharmaceutical manufacturing, special equipment manufacturing, computer, communication, and other electronic equipment manufacturing, railway, shipping, aerospace, and other transportation equipment manufacturing, and automobile manufacturing. Manufacturing enterprises in Beijing tend to high-tech and high value-added industries. On the other hand, Tianjin is dominated by special equipment manufacturing, pharmaceutical manufacturing, automobile manufacturing, electrical machinery and equipment manufacturing. However, the development of technology-intensive enterprise clusters in Hebei Province is still insufficient. Even though Baoding, Shijiazhuang, Tangshan, Cangzhou have developed to some extent, there is still an evident gap between them and Beijing and Tianjin. Baoding City, known for the production of automobiles, chemical raw materials and products, and electrical machinery and equipment, has managed to form a relatively complete industrial chain of vehicle manufacturing. Furthermore, Shijiazhuang'g pharmaceutical industry has a strong market influence in Hebei Province. Due to their port advantages, Caofeidian District in Tangshan City and Huanghua District in Cangzhou City are key areas for heavy chemical enterprises, manufacturing chemical raw materials and products, and special and general equipment in Hebei Province.

The number of resource-intensive enterprises in the Beijing-Tianjin-Hebei region ranks second among the four types of enterprises. These enterprises are predominantly distributed

Table 1. Number of the listed manufacturing enterprises by type.

| Type (quantity/piece) | Industry (quantity/unit) |
|---|---|
| **Technology-intensive (354)** | Electrical machinery and equipment manufacturing industry (49), chemical raw materials and chemical products manufacturing industry (41), computer, communication and other electronic equipment manufacturing industry (50), automobile manufacturing industry (39), railway, ship, aerospace and other transportation equipment manufacturing industry (24), general equipment manufacturing industry (20), pharmaceutical manufacturing industry (68), instrument manufacturing industry (7), and special equipment manufacturing industry (56) |
| **Labor-intensive (63)** | Textile and garment industry (4), textile industry (2), furniture manufacturing industry (2), wine, beverage and refined tea manufacturing industry (12), agricultural and sideline food processing industry (13), leather, fur, feather and their products and footwear industry (3), food manufacturing industry (15), and printing and recording media reproduction industry (12) |
| **Capital-intensive (27)** | Ferrous metal smelting and rolling processing industry (11), wood processing and wood, bamboo, rattan, palm and grass products industry (1), and nonferrous metal smelting and rolling processing industry (15) |
| **Resource-intensive (161)** | Non-metallic mineral products industry (82), comprehensive utilization of waste resources (6), metal products industry (40), other manufacturing industry (5), petroleum processing, coking and nuclear fuel processing industry (7), rubber and plastic products industry (17), and paper and paper products industry (4) |

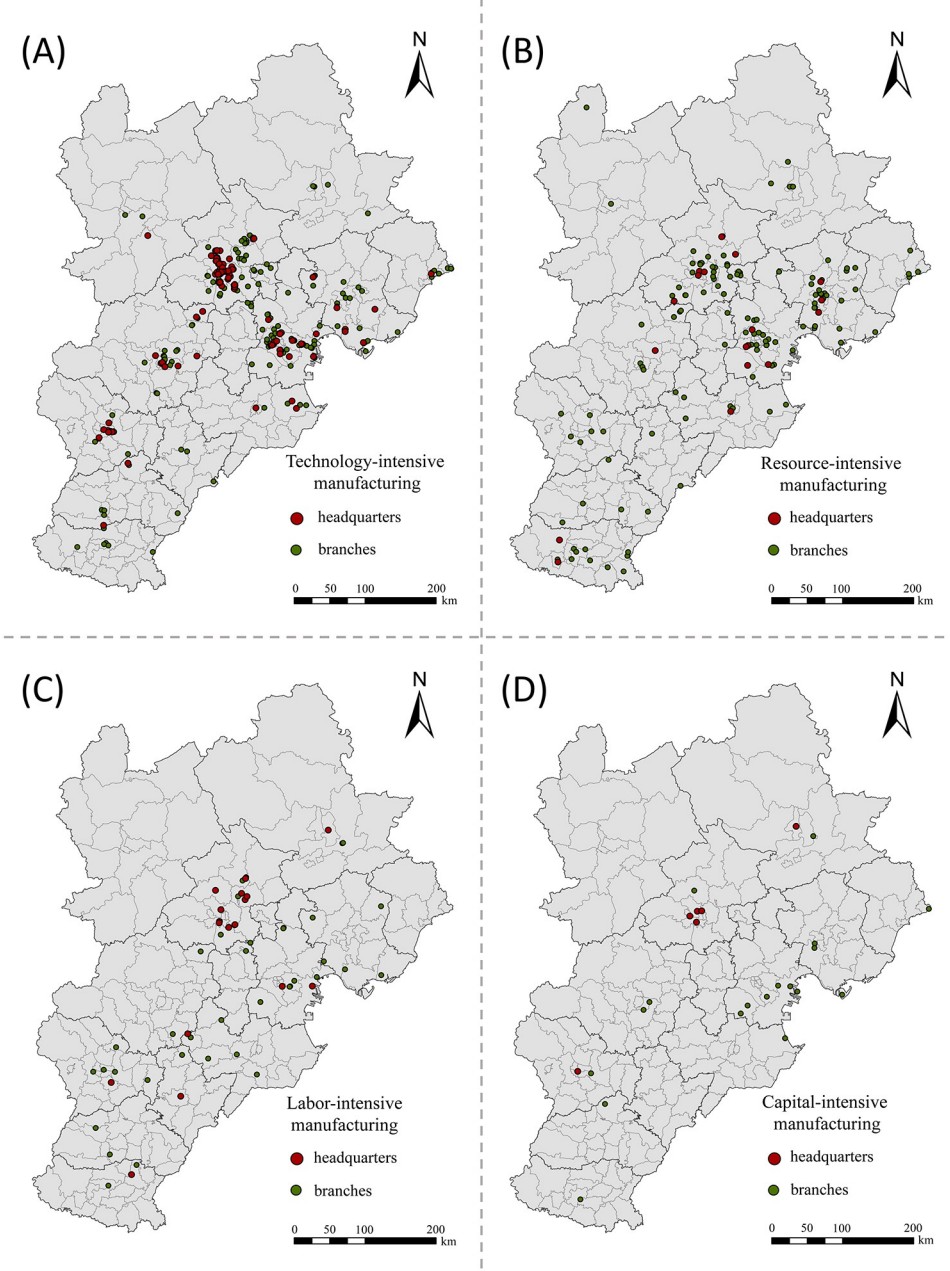

**Fig 3. Spatial distribution of the classified listed manufacturing enterprises in the Beijing-Tianjin-Hebei region.**
(A) Technology-intensive manufacturing; (B) Resource-intensive manufacturing; (C) Labor-intensive manufacturing;
and (D) Capital-intensive manufacturing.

in Beijing, Tianjin, Handan, and Tangshan, where the predominant industries are related to
metal products and non-metallic mineral products. In addition to Beijing and Tianjin, the
number of labor-intensive enterprises in central and Southern Hebei is also relatively large.
Conversely, the number of capital-intensive enterprises in Beijing, Tianjin, and Hebei is small
and scattered across the region.

**Table 2. Out-degree and in-degree of the city-level nodes in the Beijing-Tianjin-Hebei region.**

| Rank | City | Out-degree | Rank | City | In-degree |
|------|------|------------|------|------|-----------|
| 1 | Beijing | 322 | 1 | Tianjin | 103 |
| 2 | Tianjin | 41 | 2 | Tangshan City | 88 |
| 3 | Baoding City | 38 | 3 | Langfang City | 47 |
| 4 | Shijiazhuang City | 35 | 4 | Cangzhou City | 35 |
| 5 | Tangshan City | 21 | 5 | Baoding City | 34 |
| 6 | Handan City | 9 | 6 | Qinhuangdao City | 31 |
| 7 | Xingtai City | 6 | 7 | Shijiazhuang City | 31 |
| 8 | Chengde City | 6 | 8 | Beijing | 30 |
| 9 | Hengshui City | 4 | 9 | Chengde City | 25 |
| 10 | Zhangjiakou City | 4 | 10 | Xingtai City | 20 |
| 11 | Cangzhou City | 3 | 11 | Handan City | 19 |
| 12 | Qinhuangdao City | 0 | 12 | Hengshui City | 16 |
| 13 | Beijing | 0 | 13 | Zhangjiakou City | 10 |

## Analysis of spatial structure characteristics of the Beijing-Tianjin-Hebei urban network

Based on the data gathered from the headquarters and branches of 605 listed manufacturing enterprises, this paper constructs a 199×199 directed multivalued network matrix at the county scale. In the matrix, the vertical coordinate represents the output place of the city where the manufacturing headquarters are located, while the horizontal coordinate represents the receiving place of the city where the manufacturing branches are located. The spatial structure of the urban network in the Beijing-Tianjin-Hebei region is analyzed by UCINET 6.

### Characteristics of urban network nodes

The radiation and agglomeration capacity of different regions of the Beijing-Tianjin-Hebei region are significantly different (Tables 2 and 3). There are 46 districts and counties with a

**Table 3. Out-degree and in-degree of the county-level nodes in Beijing-Tianjin-Hebei region.**

| Rank | Country | Out-degree | Rank | Country | In-degree |
|------|---------|------------|------|---------|-----------|
| 1 | Haidian District (Beijing) | 139 | 1 | Dongli District (Tianjin) | 46 |
| 2 | Dongcheng District (Beijing) | 124 | 2 | Binhai New Area (Tianjin) | 33 |
| 3 | Daxing District (Beijing) | 50 | 3 | Gaocheng District (Shijiazhuang) | 29 |
| 4 | Chaoyang District (Beijing), Chang'an District (Shijiazhuang) | 39 | 4 | Daxing District (Beijing) | 28 |
| 5 | Changping District (Beijing) | 29.5 | 5 | Shunyi District (Beijing) | 26 |
| 6 | Fengtai District (Beijing), Lianchi District (Baoding) | 25 | 6 | Changping District (Beijing) | 24 |
| 7 | Zhuozhou (Baoding), Fengrun District (Tangshan) | 22 | 7 | Caofeidian District (Tangshan) | 23 |
| 8 | Qiaoxi District (Shijiazhuang) | 21 | 8 | Jingxiu District (Baoding) | 22 |
| 9 | Lunan District (Tangshan), Shunyi District (Beijing) | 18 | 9 | Huanghua District (Cangzhou) | 20 |
| 10 | Xiqing District (Tianjin) | 17 | 10 | Haigang District (Qinhuangdao) | 19 |
| 11 | Jinghai District (Tianjin) | 15 | 11 | Fengrun District (Tangshan) | 18 |
| 12 | Ningjin County (Xingtai), Shijingshan District (Beijing), and Binhai New Area (Tianjin) | 14 | 12 | Fengnan District (Tangshan) | 17 |
| 13 | Wu'an County (Handan) | 11 | 13 | Huairou District (Beijing), Baodi District (Tianjin) | 16 |
| 14 | Yuhua District (Shijiazhuang), Dongli District (Tianjin) | 10 | 14 | Miyun District (Beijing) | 15 |
| 15 | Fengfeng mining area (Handan), Beichen District (Tianjin) | 9 | 15 | Fangshan District (Beijing), Dachang Hui Autonomous County (Langfang) | 14 |

radiation function in the Beijing-Tianjin-Hebei region. As they are administrative centers, Beijing, Tianjin, and Shijiazhuang have significantly higher radiation than other parts of the region. Beijing has an absolute advantage in the urban network, as its radiation range covers all cities in the region. With the exception of Mentougou District and Xicheng District, all remaining districts and counties affect the regional output network. Its radiation direction is mainly towards Tianjin and Tangshan. In addition to Baodi and Ninghe District, all the remaining districts and counties in Tianjin also exhibit a strong radiation effect. Shijiazhuang as the provincial capital city, its internal Chang'an District, Qiaoxi District, and Yuhua District firmly control the resources in the surrounding districts and counties. Furthermore, some other districts and counties, such as Fengrun District, are shown to exhibit strong control in the Beijing-Tianjin-Hebei Urban Network. Conversely, districts and counties in Qinhuangdao and Langfang have no radiation capacity.

Although the 95 districts and counties in the Beijing-Tianjin-Hebei region exhibit agglomeration capacity, most of them still have small in-degree values. The counties with strong agglomeration ability are Dongli District and Binhai New Area in Tianjin, Gaocheng District in Shijiazhuang City, Daxing District, Shunyi District and Changping District in Beijing, Caofeidian District, Fengrun District and Fengnan District in Tangshan City, Jingxiu District in Baoding City, Huanghua District in Cangzhou, and Haigang District in Qinhuangdao City. As they are adjacent to a central city or a port, these districts and counties are able to attract enterprises and thus have a strong impact on the overall access network. Dachang Hui Autonomous County of Langfang City is situated between Beijing and Tianjin, while the seaport area of Qinhuangdao City has port resources, meaning that both districts have a certain resource agglomeration ability. Within the Beijing-Tianjin-Hebei region, more than half of the counties have an in-degree value of 0, indicating that the radiation effect of the central districts and counties is limited, with most of the marginal districts and counties unable to accept the radiation drive.

## Overall network characteristics

**Overall network density characteristics.** Based on multi-valued network calculations, the urban network density in the Beijing-Tianjin-Hebei region is determined to be low (0.0207 in 2020). The number of counties in this region is 199, while the maximum number of possible connections is 39,601. However, in reality, there are only 217 connections, suggesting that the urban network development based on the listed manufacturing enterprises is low and that the connection between cities is weak.

**Characteristics of the overall network connection strength.** Based on the data obtained from headquarters and branches, this paper extracts the urban connection pairs between the prefecture level and the district/county scale. It goes further towards classifying urban connection strength and node strength using the natural breakpoint method in ArcGIS 10.2. As a result, the paper generates the urban network spatial grid of the Beijing-Tianjin-Hebei region. Even though all prefecture-level cities in this region were part of the regional network connection, there are still differences in the connection degree between them (Fig 4). Beijing is closely connected to most cities in the region and has the most high-level links. Conversely, the main links between cities in Hebei Province are at the medium and low levels.

At the county level (Fig 5), 107 districts and counties were part of the network connection. The main network links are found in Haidian District, Dongcheng District, Daxing District, Changping District, Shunyi District and Chaoyang District in Beijing, Dongli District and Binhai New Area in Tianjin, and Fengrun District in Tangshan City. Certain districts and counties of Shijiazhuang, Baoding and Qinhuangdao City also have multiple links. Handan City,

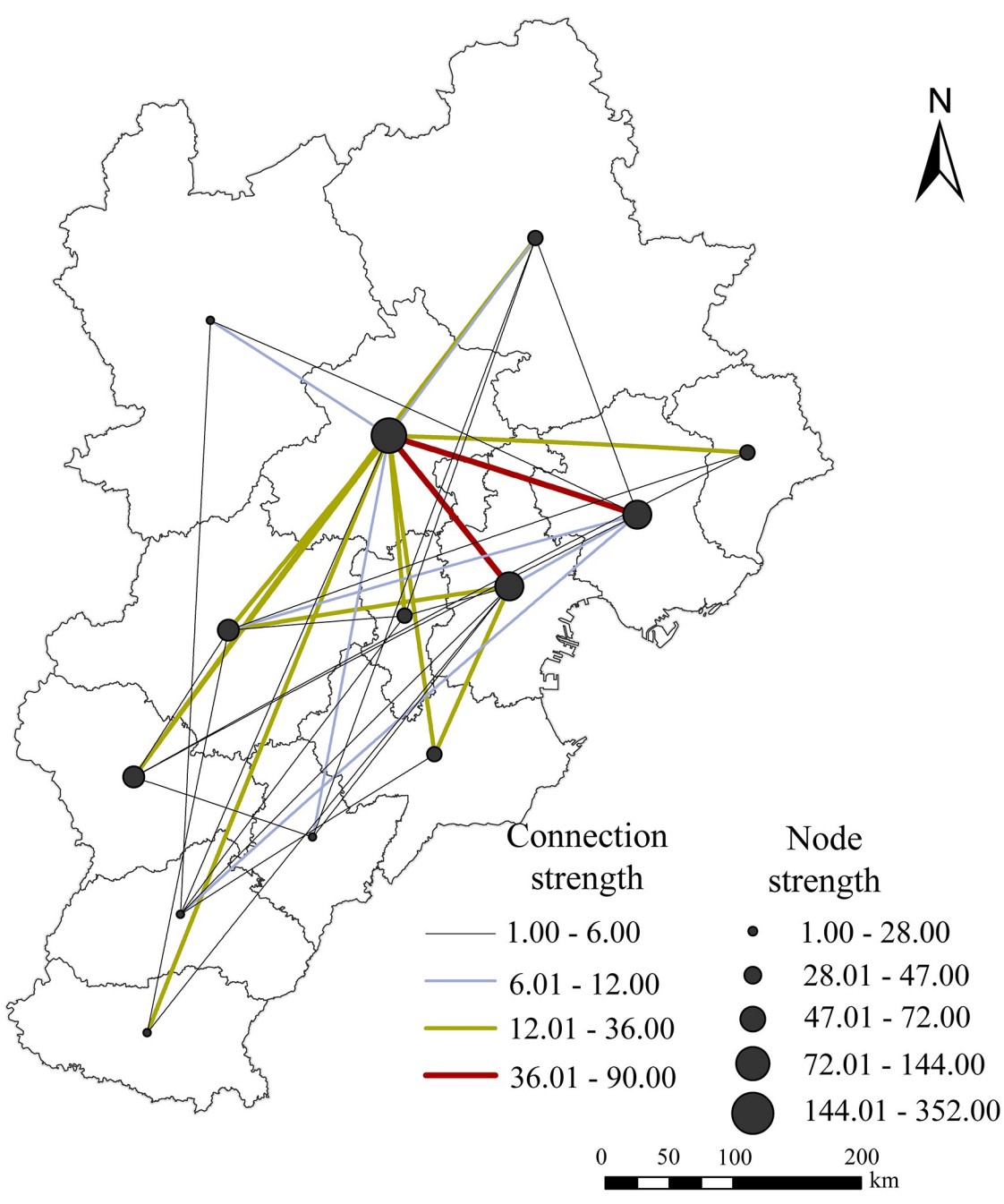

**Fig 4. Spatial pattern of the urban network at the prefecture level.**

Xingtai City, Cangzhou City, Hengshui City, Langfang City, Zhangjiakou City, and Chengde City have fewer internal districts and counties network connection lines, and many districts and counties are not involved in network construction, so they become areas with inactive urban network flow.

Compared to the polycentric structure with Shanghai, Nanjing and Hangzhou as the core, and high-level network connections represented by the Yangtze River Delta manufacturing

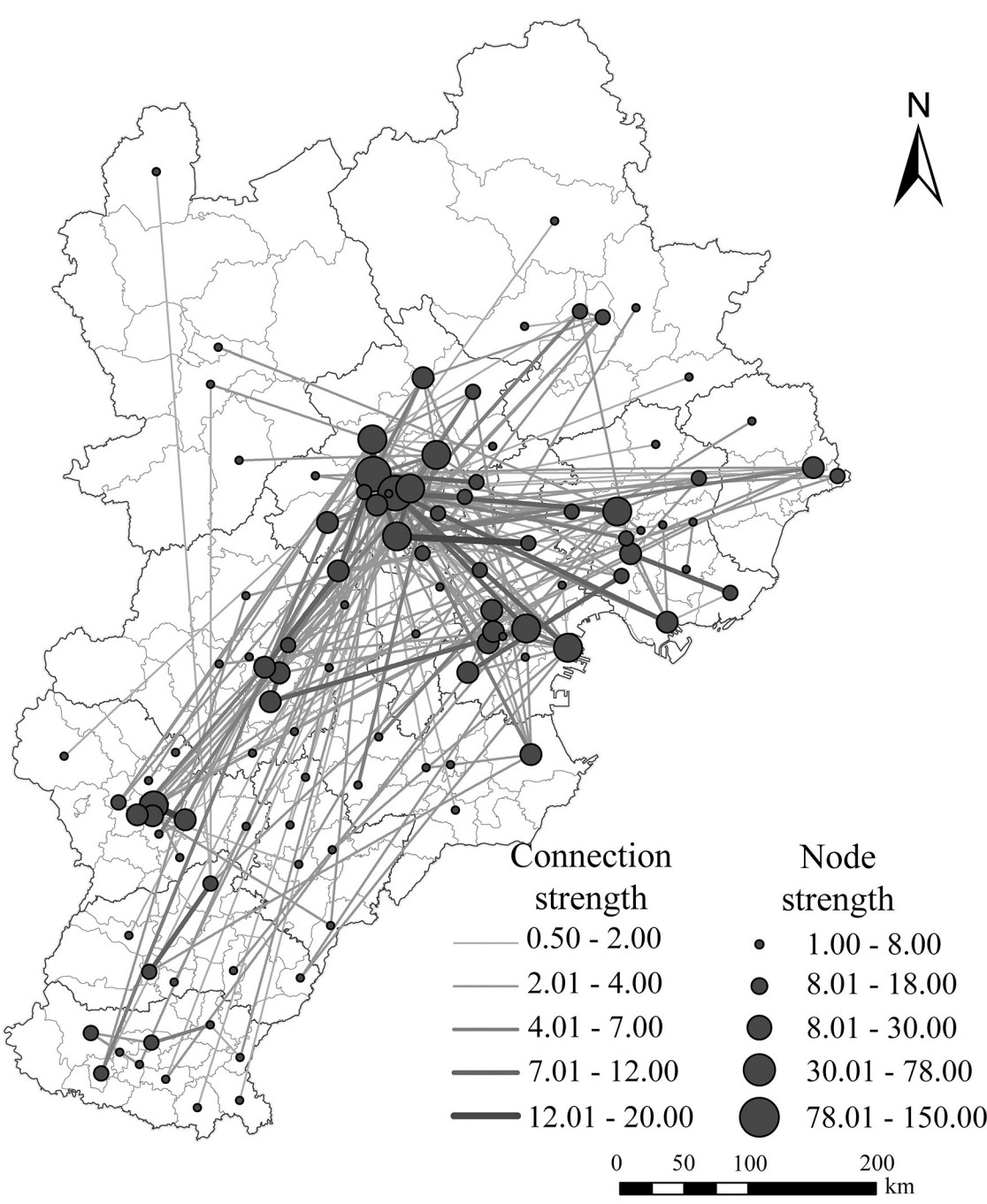

**Fig 5. Spatial pattern of the urban network at the county level.**

enterprise network [39], the urban network linkages in the Beijing-Tianjin-Hebei region are dominated by the dual-core structure and the low to medium level linkages. The core of urban network development is the connection between Beijing and Tianjin, with Beijing-Tianjin-Tangshan and Beijing-Baoding-Shijiazhuang forming the main axis, showing a "dense in the southeast and sparse in the northwest" pattern.

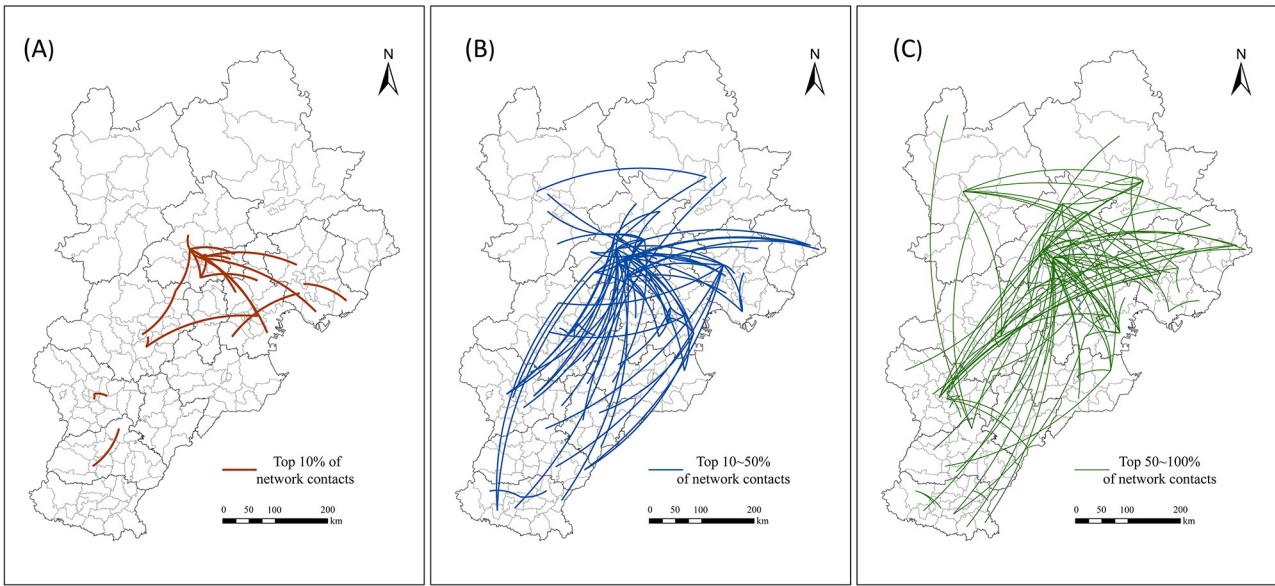

**Fig 6. Hierarchical pattern of the urban network at the county level.** (A) Top 10% of network contacts; (B) Top 10~50% of network contacts; and (C) Top 50~100% of network contacts.

### Hierarchical network characteristics

To further reveal the spatial hierarchical characteristics of the Beijing-Tianjin-Hebei urban network, the connection strengths between districts and counties in the network are ranked from high to low. The top 10%, 10~50%, and 50~100% network linkage pairs are extracted, and a spatial hierarchical network map was drawn using ArcGis (Fig 6).

The number of contact pairs in the top 10% of the urban network is relatively small. They are primarily located between the districts and counties in Beijing, Tianjin, Shijiazhuang, and their adjacent districts and counties. Beijing and Tianjin with high intensity network links form the core area of the entire network. Compared to the connection lines of the top 10% of urban networks, the top 10% to 50% of the network connections have increased significantly. In addition to neighboring cities, long-distance connection lines also increased significantly, but they were mainly concentrated in the southeast of the region and relied on the network connection of high-level cities. In addition to the great strength of the county network in Beijing and Tianjin, Qiaoxi District in Shijiazhuang City, Huanghua District of Cangzhou City, Caofeidian District in Tangshan City and Haigang District in Qinhuangdao City have also formed a strong local network. However, the network in the northwest of the region is developing slowly. Only Shuangluan District and Shuangqiao District in Chengde City are connected to the southeastern part of the region. When the network structure increases from the top 50% to the total network, long-distance weak connection lines appear, and the overall distribution characteristics of the network do not change significantly. The connections are still concentrated in the southeastern portion of the region, while the network integration degree in the northwest remains low.

The hierarchical network analysis further confirms that Beijing and Tianjin are at the core of the Beijing-Tianjin-Hebei city network and form the most stable and highly connected hierarchy, while the cities in Hebei province are at a subordinate position in the network with intermediate force capable of increasing the strength of connections in the city network

hierarchy. This finding is consistent with the findings of Yu Qian [40] who studied the network structure within Beijing-Tianjin-Hebei based on data on inter-firm fund transactions.

## Cyberspace structure of the listed manufacturing enterprises by type

The contact network of different types of manufacturing enterprises (technology-intensive, resource-intensive, labor-intensive and capital-intensive) is complex and diverse. This paper uses ChordDiagram to visualize the topological relationship between counties in the Beijing-Tianjin-Hebei region. An arc in a chord diagram corresponds to a network node. The larger the arc corresponding to the central angle and the longer the arc length, the higher the degree index of the network node. The connection between arcs reflects the topological relationship between different nodes. The larger the connection width, the stronger the correlation between the nodes (Fig 7).

The technology-intensive enterprise connections are mostly internal linkages of districts and counties under the jurisdiction of each city. Haidian, Chaoyang and Changping Districts in Beijing, Chang'an District in Shijiazhuang, Lianchi and Zhuozhou Districts in Baoding, and Nankai District in Tianjin have high centrality in the technology-intensive enterprise network.

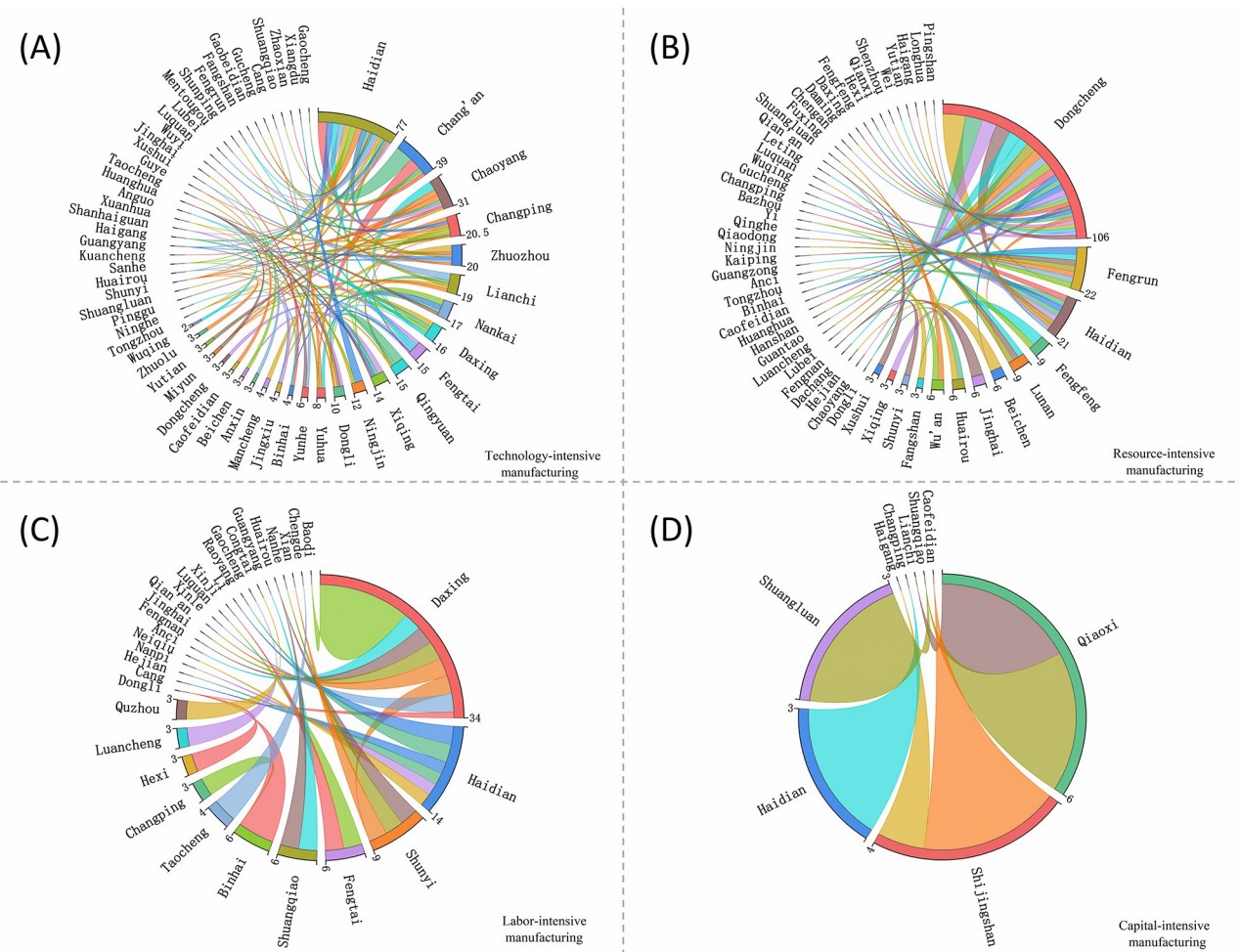

**Fig 7. Spatial distribution of the urban network of the listed manufacturing enterprises by type.** (A) Technology-intensive manufacturing; (B) Resource-intensive manufacturing; (C) Labor-intensive manufacturing; and (D) Capital-intensive manufacturing.

As the intensity of connections decreases, the number of connections in the technology-intensive enterprise network increases and the range of districts and counties expands.

The number of district and county units involved in the resource-intensive enterprise network is less than that of the technology-intensive enterprise network. Dongcheng District in Beijing has a central position in this network. Dongcheng District of Beijing has a core position in this network. It has a strong connection with Chaoyang District, Haidian District of Beijing and Dongli District of Tianjin.. The number of district and county units involved in the labor-intensive enterprise network further decreases, and high-grade linkage routes are rare, which shows that the overall network linkage structure is centered on Beijing's Daxing District and Haidian District. The capital-intensive enterprise network involves the smallest number of district and county units, with a small number of lines and a low degree of network development.

The development of the four types of manufacturing networks differs significantly, with more high-grade linkage routes in the networks of technology-intensive and resource-intensive enterprises, and mostly low-grade linkage routes in the networks of labor-intensive and capital-intensive enterprises. Except for the capital-intensive manufacturing network that does not form an obvious core area, the other three types of manufacturing networks are mainly centered on districts and counties within Beijing in order to connect to the surrounding and distant areas.

## Agglomerative subgroup analysis of urban network

In order to analyze the urban network clusters and agglomeration in the Beijing-Tianjin-Hebei region, this paper uses convergence of iterated correlation (CONOR) in UCINET. With it, the paper divides the agglomerative subgroups into five categories (Fig 8) and calculates their density matrix (Table 4).

Subgroup 1 consists of 27 districts and counties, including Chang'an District, Qiaoxi District and Lianchi District in Baoding City. The distribution of subgroup members is relatively scattered, and the degree of contact is low. Externally, they have a high connection only with Subgroup 3. Subgroup 2 consists of 28 economically developed districts and counties, including Haidian, Dongcheng, and Chaoyang Districts in Beijing. This subgroup is the largest in the region, as it includes 232 enterprises, mostly technology-intensive and resource-intensive, such as the non-metallic mineral products industry, computer, communication and other electronic equipment manufacturing industry, and special equipment manufacturing industry. Subgroup 2 shows a small range of agglomeration in space. Compared to the other subgroups, Subgroup 2 shows the highest degree of internal connectivity. It is also closely related to other subgroups, exhibiting a strong ability to control and absorb external resources. Subgroup 3 consists of 24 districts and counties, including Gaocheng District in Shijiazhuang City and Jinghai District in Tianjin City. Due to the lack of a driving role of core cities, the internal and external connection density of Subgroup 3 is low.. Subgroup 4 is composed of 27 districts and counties, including Daxing District, Changping District, Shunyi District in Beijing, Xiqing District, Binhai New Area in Tianjin and Zhuozhou District of Baoding City. There is a total of 213 diverse enterprises within this subgroup. The spatial agglomeration of Subgroup 4 is evident, while its internal connections are high. There was no relationship observed between Subgroup 5 and any other subgroup.

Agglomerative subgroups of the entire region form an outward diffusion trend, with Subgroups 2 and 4 constituting the core. These two subgroups are closely related and have strong economic relevance. On the other hand, Subgroups 1 and 3 are located on the secondary edge and are closely related to the core subgroups, receiving their radiation and driving effect.

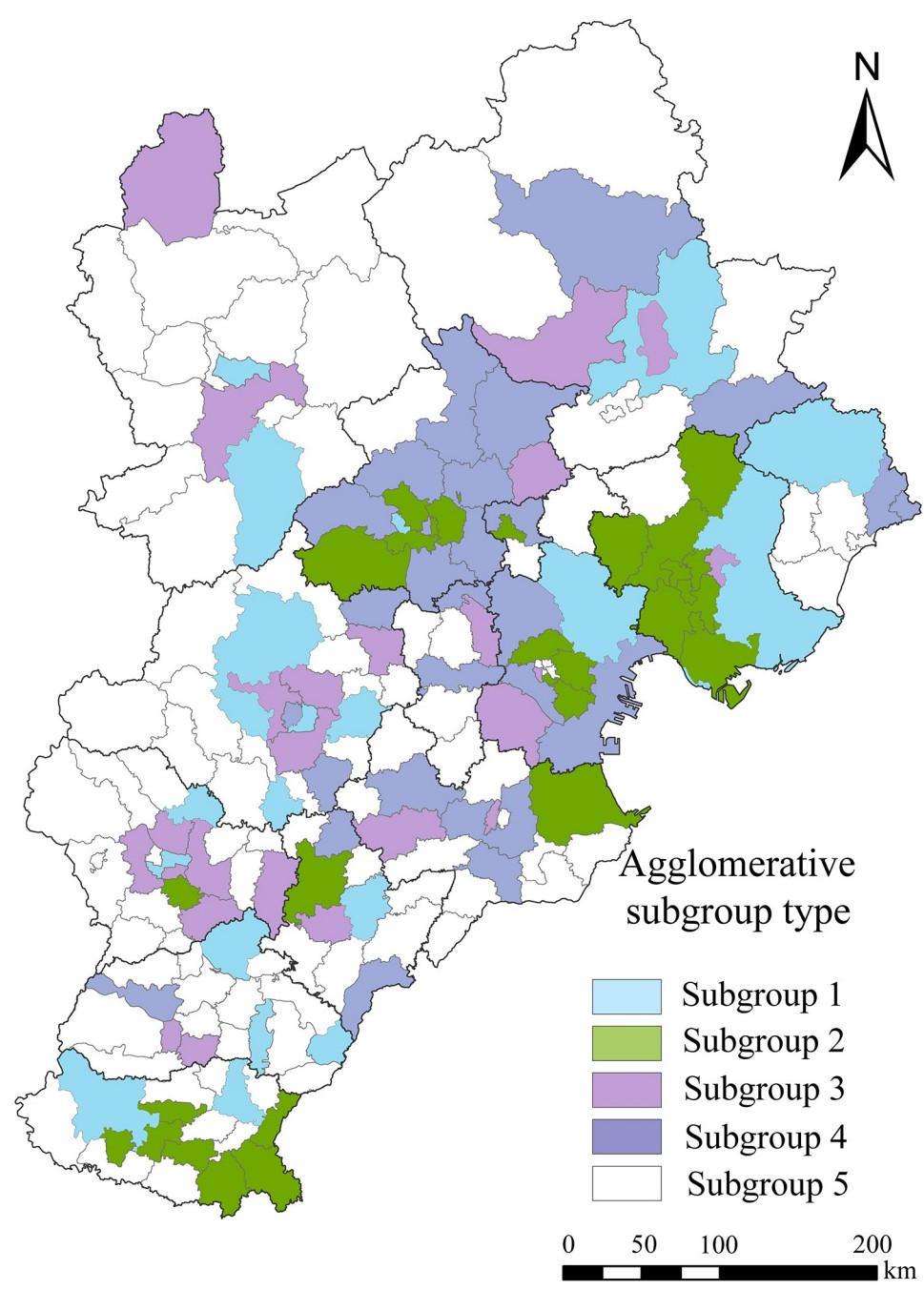

**Fig 8. Agglomeration subgroup map of urban network.**

**Table 4. Density matrix of agglomerative subgroups of the urban network in the Beijing-Tianjin-Hebei region.**

| Agglomerative subgroup | I | II | III | IV | V |
|---|---|---|---|---|---|
| I | 0.036 | 0.036 | 0.113 | 0.027 | 0.000 |
| II | 0.052 | 0.217 | 0.040 | 0.249 | 0.000 |
| III | 0.014 | 0.048 | 0.009 | 0.057 | 0.000 |
| IV | 0.047 | 0.048 | 0.060 | 0.085 | 0.000 |
| V | 0.000 | 0.000 | 0.000 | 0.000 | 0.000 |

Table 5. Index selection of influencing factors of the urban network in the Beijing-Tianjin-Hebei region.

| Dependent variable (Y) | Independent variable (X) | Independent variable interpretation |
|---|---|---|
| Beijing-Tianjin-Hebei Urban Network incidence matrix | Administrative relations | Administrative boundary 0–1 network ($X_1$) |
| | Geospatial distance | Geospatial distance network ($X_2$) |
| | City size differences | County population difference network ($X_3$) |
| | Economic developmental differences | GDP per capita difference network ($X_4$) |
| | Industrial structure differences | Difference network of the proportion of secondary industry in GDP ($X_5$) |
| | Labor cost differences | On-the-job worker average wage difference network ($X_6$) |

Compared to other subgroups, the economic development of the inner districts and counties of Subgroup 5 is relatively backward, which is why it is on the edge of the network.

## Analysis of the influencing factors of the Beijing-Tianjin-Hebei urban network

### Selection of influencing factors and model construction

Existing research argues that the formation of an urban network depends on several factors [35, 38]. In order to determine the influencing factors of the spatial structure of the urban network in the Beijing-Tianjin-Hebei region, this paper analyzes the administrative relationship, spatial distance, city size, economic development, industrial structure, and labor cost difference within this region (Table 5). The measurement model was constructed as follows:

$$Y = f(x_1, x_2, x_3, x_4, x_5, x_6) \tag{4}$$

In Eq (4), $Y$ represents the dependent variable of the Beijing-Tianjin-Hebei urban network relationship matrix, while $X_1$-$X_6$ represents the independent variable. This is explained as follows:

1. $X_1$: Administrative relationship denotes whether two districts or counties are within the same administrative division. Districts and counties under the same prefecture-level city can avoid the production division problem caused by administrative division. In this paper, dummy variables 1 and 0 are introduced. Districts and counties under the same prefecture-level city are assigned the value 1, while those belonging to different prefecture-level cities are assigned the value 0.

2. $X_2$: Geospatial distance significantly affects the spatial growth of multi-sector enterprises. In general, the spatial expansion of enterprises follows a pattern of near expansion. Location proximity is thus more conducive for the information exchange and contact between enterprises. As the distance between enterprises increases, the strength of inter-enterprise ties gradually weakens. This paper used the point distance tool in ArcGIS 10.2 to measure the spatial distance between different districts and counties, and then constructs the spatial distance relationship matrix $X_2$.

3. $X_3$: City size difference. High-level cities tend to have greater power resources and economic efficiency, which increases population inflow, enhances market demand, and brings more enterprises. Therefore, this paper constructs a city-level difference matrix $X_3$ based on the county population difference of each two districts and counties.

4. $X_4$: Difference in the level of economic development. In general, regions with better economic development, better infrastructure, transportation, and regional policy dividend will have a strong attraction for listed manufacturing enterprises. The relationship matrix $X_4$ of the GDP per capita difference is constructed based on the difference between the GDPs per capita of two districts or counties.

5. $X_5$: Differences in industrial structure. The location of the listed manufacturing enterprises is closely related to the local industrial structure. The relationship matrix X5 of industrial structure difference is constructed by the difference of the proportion of the secondary industry output value to the regional GDP between two districts or counties.

6. $X_6$: Labor cost difference. In order to generate higher profits, enterprises tend to invest in areas with low labor costs. The labor cost difference matrix $X_6$ is constructed based on the difference between the average wage of on-the-job workers between two districts or counties.

The data used in these calculations are obtained from the 2019 Beijing regional statistical yearbook, the Tianjin statistical yearbook, the Hebei Economic Yearbook, the China county statistical yearbook, relevant municipal statistical yearbooks, and the County national economic and social development statistical bulletin. Furthermore, administrative relations and geographical distance themselves can constitute relational matrices. In order to construct the corresponding difference matrix and calculate the absolute difference, the remaining influencing factors take the data difference between two counties as an index and take the absolute value of the actual difference between the districts and counties. In order to eliminate the influence of dimension, the difference matrix of the mentioned explanatory variables was standardized, after which the QAP analysis was conducted.

## QAP analysis

**QAP correlation analysis.** The QAP correlation analysis examines whether two variables are related. In general, its purpose is to test the correlation between different types of urban attribute relationship matrix and urban network relationship matrix. The paper entered the binary relation matrix of the Beijing-Tianjin-Hebei Urban Network and other independent variable relation matrices (obtained by binarization with the average value as the cut-off value) into the UCINET software and selected 5000 random permutations. Table 6 provides the analysis results.

**Table 6. QAP correlation analysis of urban network and influencing factors in the Beijing-Tianjin-Hebei region.**

| Influence factor | Correlation coefficient | Standard deviation | Minimum value | Maximum value |
|---|---|---|---|---|
| Administrative relation matrix | 0.076*** | 0.005 | -0.015 | 0.021 |
| Geospatial distance matrix | -0.036*** | 0.008 | -0.026 | 0.027 |
| City size difference matrix | 0.028*** | 0.010 | -0.029 | 0.032 |
| Difference matrix of economic development level | 0.044*** | 0.011 | -0.033 | 0.041 |
| Industrial structure difference matrix | 0.018** | 0.009 | -0.027 | 0.029 |
| Labor cost difference matrix | 0.039*** | 0.010 | -0.033 | 0.036 |

Note:

* * * indicates that the significance test is less than 0.01

* * indicates that the significance is less than 0.05

* indicates that the significance is less than 0.1.

Correlation analysis of administrative relationship, spatial distance, city size, level of economic development, labor cost difference and urban network passed the 1% significance test. Moreover, the difference in industrial structure passed the 5% significance test, showing that these seven factors significantly correlate with the network structure of Beijing, Tianjin, and Hebei. Among them, the correlation coefficient between spatial distance and urban network was -0.036. This indicates that the spatial distance between districts and counties has a negative effect on the formation of the urban network.

**QAP regression analysis.** The QAP regression analysis studies the regression relationship between multiple variables and a single variable. Its purpose is to identify the influence of different types of urban attribute relationship matrix on the urban network relationship matrix. In this paper, the regression analysis was carried out based on the mentioned correlation analysis. The analysis results are provided in Table 7.

The regression analysis illustrated that the administrative relationship has the most significant impact on the Beijing-Tianjin-Hebei Urban Network, with a regression coefficient of 0.0211. This observation indicates that the connections between districts and counties situated in the same prefecture-level city are stronger than those belonging to different prefecture-level cities within the urban. This is due to the fact that certain policies implemented by local governments restrict the cross-regional flow of production factors, resulting in closer economic relations between districts and counties in the same administrative division.

Second, the regression coefficient of spatial distance was calculated to be -0.0022. This suggests that distance between cities negatively affects the formation of urban networks. Furthermore, it indicates that spatial proximity is more conducive to the communication of manufacturing enterprises between districts and counties. Lastly, this shows that strengthening the transportation network promotes the coordinated development of Beijing, Tianjin, and Hebei.

Third, the difference in city size passed the 5% significance test, showing that cities with large population inflows, market demand, and more public resources are more attractive to enterprises. In this case, the regression coefficient is positive, indicating that cities with larger differences in size are more likely to connect with each other.

Fourth, the difference in the level of economic development is significantly positive at the 1% level. Due to their superior conditions, regions with a high level of economic development have a great impact on the surrounding low-level regions, driving their economic

**Table 7. QAP regression analysis of urban network and influencing factors in the Beijing-Tianjin-Hebei region.**

| Influence factor | Non-standard parameters | Standard parameters | P-value |
|---|---|---|---|
| Intercept term | -0.0001 | 0.0000 | - |
| Administrative relation matrix | 0.0211*** | 0.0773 | 0.0000 |
| Geospatial distance matrix | -0.0022** | -0.0147 | 0.0200 |
| City size difference matrix | 0.0030** | 0.0196 | 0.0110 |
| Difference matrix of economic development level | 0.0053*** | 0.0331 | 0.0010 |
| Industrial structure difference matrix | 0.0015* | 0.0102 | 0.1000 |
| Labor cost difference matrix | 0.0045*** | 0.0293 | 0.0020 |
| Number of samples | 39800 | | |

Note

* * * indicates that the significance test is less than 0.01

* * indicates that the significance is less than 0.05

* indicates that the significance is less than 0.1.

development. Therefore, urban networks based on manufacturing enterprises tend to connect in regions with large differences in their economic development.

Fifth, the coefficient of regional industrial structure difference is positive. Suggesting that a greater industrial structure gap enables greater contact within region. The difference in regional industrial structure shows that each region has different types of resources. In order to utilize the complementary advantages of resources, the regions will strengthen the links between each other.

Sixth, labor cost greatly impacts the location and operation costs of enterprises. A positive regression labor cost coefficient suggests that districts and counties with a larger gap in labor cost are more likely to be connected, while high-cost areas are more likely to exhibit a radiation effect. Low-cost areas are crucial because the listed manufacturing enterprises establish branches in them.

## Discussion and conclusion

This paper constructed a subordinate connection model based on the data of headquarters and branches of the listed manufacturing enterprises. It employed the social network analysis method to determine the spatial structure and influencing factors of the urban network of the Beijing-Tianjin-Hebei region. Based on this, the following conclusions were drawn:

1. The resource allocation in the Beijing-Tianjin-Hebei region is unbalanced, while the difference in urban radiation and agglomeration capacity is significant. Beijing, Tianjin and Shijiazhuang are administrative central cities and therefore have shown a strong resource allocation capacity. On the other hand, Baoding, Tangshan, and their inner districts and counties do not constitute administrative central cities, but they have a greater resource allocation and agglomeration capacity within the urban network.

2. The level of development of the urban network within the Beijing-Tianjin-Hebei region is relatively low, consisting mainly of middle and low-level connections. Beijing and Tianjin represent the strongest centers in the network, while Shijiazhuang, Tangshan and Baoding constitute the stronger secondary network centers. All the aforementioned cities constitute the core framework for the development of the regional urban network, with the overall network connectivity forming a "dense in the southeast and sparse in the northwest" pattern. Different levels of network links exhibit spatial heterogeneity, and the number of high-level city network links is small, noticed mainly in Beijing, Tianjin, Shijiazhuang, and their adjacent districts and counties. As the connection level decreases, the network connection between remote districts and counties experiences a significant increase, mostly concentrated in the southeastern part of the region. The network development of the listed manufacturing enterprises differs significantly by type. The spatial scope and number of links involved in the network of technology-intensive enterprises are much larger than those of the other three types of enterprises. Except for capital-intensive enterprises, the other three types of manufacturing networks are mainly connected to the surrounding and remote areas with the internal districts and counties of Beijing as the center.

3. The inner and outer degree of contact between the core subgroups of the Beijing-Tianjin-Hebei Urban Network is high, with the subgroup members close to each other at the regional level. The members of the sub-marginal subgroup exhibit a low level of economic development and weak cohesion, mainly driven by the radiation of the core subgroups. Furthermore, marginal subgroups form few relationships with other subgroups in the region. In sum, the relationship between and within subgroups should be improved.

4. The results of the QAP analysis suggest that administrative divisions more closely connect districts and counties under the same prefecture-level city. In terms of geographical distance, the region should work on improving regional traffic links to strengthen the relationship between districts and counties, and thus promote the development of its urban network. Areas with high levels of urban size and economic development will radiate into the surrounding low level areas, thus promoting balanced regional development. Furthermore, differences in industrial structure may promote the complementary advantages of different types of resource elements among regions. Lastly, the listed manufacturing enterprises tend to open branches in areas with low labor costs, which provides favorable conditions for regional ties.

This paper analyzes the characteristics and influencing factors on the urban network structure in the Beijing-Tianjin-Hebei region based on the listed manufacturing enterprises at the district and county scales, which are complementary to the national, provincial and prefecture-level city scales. However, while the county network is formed by the joint action of several factor flows, the number of listed manufacturing enterprises in the region is limited. As the representative network type is relatively unique, there are still a number of issues in describing the development of the county network. Therefore, future research should conduct more in-depth studies of the Beijing-Tianjin-Hebei County area network using different types of data. Furthermore, this study only described the spatial structure characteristics of the Beijing-Tianjin-Hebei Urban Network for one year. At present, a longitudinal comparative analysis is lacking and should be considered in further research.

## Supporting information

**S1 Fig. Data of the listed manufacturing enterprises.** The figure shows the spatial distribution of the listed manufacturing enterprise headquarters and branches in the Beijing-Tianjin-Hebei region.
(ZIP)

**S2 Fig. Data of the listed manufacturing enterprises by type.** The figure shows the spatial distribution of technology-intensive, resource-intensive, labor-intensive and capital-intensive manufacturing enterprises in the Beijing-Tianjin-Hebei region.
(ZIP)

**S3 Fig. Connection strength and node strength data.** The figure shows the spatial pattern of urban networks at the prefecture and district levels in the Beijing-Tianjin-Hebei region.
(ZIP)

**S4 Fig. Aggregate subgroup hierarchical data.** The figure shows the spatial distribution of subgroups at the district and county levels in the Beijing-Tianjin-Hebei region.
(ZIP)

**S1 Table. Data of the node out-degree and in-degree.** Table shows out-degree and in-degree values at the prefecture and district levels in the Beijing-Tianjin-Hebei region.
(XLSX)

## Acknowledgments

We sincerely thank colleagues from the Faculty of Geographical Sciences for their valuable input during the research and writing process.

## Author Contributions

**Conceptualization:** Chuning Miao.

**Data curation:** Wengang Wang.

**Formal analysis:** Wengang Wang.

**Funding acquisition:** Wengang Wang.

**Investigation:** Chuning Miao.

**Methodology:** Haihang Yu, Can Li.

**Project administration:** Chuning Miao.

**Resources:** Haihang Yu.

**Software:** Wengang Wang.

**Supervision:** Chuning Miao.

**Validation:** Can Li.

**Visualization:** Haihang Yu.

**Writing – original draft:** Wengang Wang.

**Writing – review & editing:** Chuning Miao.

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
