## [Decision Letter · Decision Letter 0]

17 Oct 2022

PONE-D-22-26849Research on the characteristics and influencing factors of Beijing - Tianjin - Hebei urban network structure from the perspective of listed manufacturing enterprisesPLOS ONE

Dear Dr. Miao,

Thank you for submitting your manuscript to PLOS ONE. After careful consideration, we feel that it has merit but does not fully meet PLOS ONE’s publication criteria as it currently stands. Therefore, we invite you to submit a revised version of the manuscript that addresses the points raised during the review process.

We look forward to receiving your revised manuscript.

Kind regards,

Jun Yang

Academic Editor

PLOS ONE

Journal Requirements:

"Science and Technology Research Fund of Hebei Normal University: Structural Evolution and Regional Collaborative Development of Beijing-Tianjin-Hebei Urban Economic Network (Project No. L2020Z07)."

3. We note that all figures in your submission contain map images which may be copyrighted. All PLOS content is published under the Creative Commons Attribution License (CC BY 4.0), which means that the manuscript, images, and Supporting Information files will be freely available online, and any third party is permitted to access, download, copy, distribute, and use these materials in any way, even commercially, with proper attribution. For these reasons, we cannot publish previously copyrighted maps or satellite images created using proprietary data, such as Google software (Google Maps, Street View, and Earth). For more information, see our copyright guidelines: http://journals.plos.org/plosone/s/licenses-and-copyright.

a. You may seek permission from the original copyright holder of all figures to publish the content specifically under the CC BY 4.0 license.  

Additional Editor Comments:

Reviewer 1

This study presents a case study takes the county-level administrative units in the Beijing-Tianjin-Hebei region as the research object, and uses the subordinate connection model and social network analysis method to analyze the spatial structure characteristics and influencing factors of the urban network in the Beijing-Tianjin-Hebei region. The topic is interesting and practical, but I recommend the authors to address the following issues:

1. It is better to supplement the analysis of spatial correlation characteristics between parent companies and subsidiaries of different types of manufacturing enterprises;

2. It is suggested to arrange the contents according to the distribution characteristics of enterprises, spatial correlation analysis, network characteristics analysis, influencing factors, etc.

Reviewer 2

This paper explores the spatial structure characteristics and influencing factors of the urban network in the Beijing-Tianjin-Hebei region, and several interesting findings are presented which have practical significance in improving the regional integration in China. Several revisions are required before moving to the further step.

1.Although the introduction section of the paper mentioned the construction of urban network by talking affiliation model, the reasons and significance of choosing listed manufacturing industry are not clearly discussed. More elaboration should be given to echo the title of this paper.

2.The dividing evidences of hierarchical network (3.3) are not clear enough. More elaboration are reqiured.

3.Although rich literature has been reviewed in this study, there is few dialogue between the empirical findings and literature. More theoretical discussion are suggested to be given in this paper, so as to highlight the theoretical contribution of this paper.

4.Professional language editing is required to express the ideas accurately.

Reviewers' comments:

Reviewer's Responses to Questions

**Comments to the Author**

1. Is the manuscript technically sound, and do the data support the conclusions?

Reviewer #1: Yes

Reviewer #2: Yes

2. Has the statistical analysis been performed appropriately and rigorously? 

Reviewer #1: Yes

Reviewer #2: I Don't Know

3. Have the authors made all data underlying the findings in their manuscript fully available?

Reviewer #1: Yes

Reviewer #2: Yes

4. Is the manuscript presented in an intelligible fashion and written in standard English?

Reviewer #1: Yes

Reviewer #2: No

5. Review Comments to the Author

Reviewer #1: This study presents a case study takes the county-level administrative units in the Beijing-Tianjin-Hebei region as the research object, and uses the subordinate connection model and social network analysis method to analyze the spatial structure characteristics and influencing factors of the urban network in the Beijing-Tianjin-Hebei region. The topic is interesting and practical, but I recommend the authors to address the following issues:

1. It is better to supplement the analysis of spatial correlation characteristics between parent companies and subsidiaries of different types of manufacturing enterprises;

2. It is suggested to arrange the contents according to the distribution characteristics of enterprises, spatial correlation analysis, network characteristics analysis, influencing factors, etc.

Reviewer #2: This paper explores the spatial structure characteristics and influencing factors of the urban network in the Beijing-Tianjin-Hebei region, and several interesting findings are presented which have practical significance in improving the regional integration in China. Several revisions are required before moving to the further step.

1.Although the introduction section of the paper mentioned the construction of urban network by talking affiliation model, the reasons and significance of choosing listed manufacturing industry are not clearly discussed. More elaboration should be given to echo the title of this paper.

2.The dividing evidences of hierarchical network (3.3) are not clear enough. More elaboration are reqiured.

3.Although rich literature has been reviewed in this study, there is few dialogue between the empirical findings and literature. More theoretical discussion are suggested to be given in this paper, so as to highlight the theoretical contribution of this paper.

4.Professional language editing is required to express the ideas accurately.

6. PLOS authors have the option to publish the peer review history of their article (what does this mean?). If published, this will include your full peer review and any attached files.

Reviewer #1: No

Reviewer #2: No

---

## [Author Response · Author response to Decision Letter 0]

26 Nov 2022

Reviewer #1:

Response 1：We thank the reviewers for raising this issue, and we have added the spatial structure of urban networks of subtypes of listed manufacturing enterprises to lines 323-351 of the manuscript, and added graphical information to facilitate a more intuitive response to the spatial association characteristics among the sub-parent companies of different types of manufacturing enterprises.

Response 2:①The distribution characteristics of enterprises are described in lines 182-230 of the manuscript.

②We believe that spatial association analysis and network characteristics are inseparable, so we describe these two parts in the chapter of "Spatial Structure Analysis of Beijing-Tianjin-Hebei City Network". The node characteristics and overall network characteristics of the city network systematically analyze the spatial association of Beijing-Tianjin-Hebei districts and counties (lines 238-295 of the manuscript), and then describe the characteristics of the city network in detail by level, type and molecular group (lines 296-379 of the manuscript).

③The final influencing factors are analyzed in lines 380-479 of the manuscript.

Reviewer #2:

Response 1: We appreciate the reviewers' careful reading of the text. Based on his/her suggestions, we have added the reasons and significance of the choice of listed manufacturing companies in the introduction section of the manuscript, in lines 88-91 of the full text.

Response 2: The issue you raised is very worthy of our consideration. The previous ranking of connection strength carries its own subjective idea and the basis of division is unclear, after referring to relevant literature and considering the number of connections in the city network of listed manufacturing enterprises in Beijing, Tianjin and Hebei, the hierarchical network is re-divided. The strength of connections between districts and counties in the network is ranked from high to low, and the top 10%, top 10%~50% and top 50%~100% network connection pairs are extracted respectively. The purpose of this division is to simplify the network analysis and facilitate a clearer description of the network characteristics at each level. For details, see lines 296-322 of the manuscript.

Response 3: Based on reviewers' suggestions, we have added a dialogue with the literature in the manuscript.

①Page 17, lines 290-293, cites Sun Zhijing's original work, which compares and analyzes manufacturing firms in the Yangtze River Delta region with those in the Beijing-Tianjin-Hebei region.

References are: 39. Sun Zhijing. Research on urban network structure in the Yangtze River Delta region based on multiple perspectives[D]. East China Normal University, 2020. https://doi.org/10.27149/d.cnki.ghdsu.2020.000422

②Page 18, lines 315-319, cites Yu Qian's findings based on inter-firm fund transaction data to study the intra-Beijing-Tianjin-Hebei network structure, which validates this paper's Beijing-Tianjin-Hebei network structure.

References are: 40.Yu Qian.Research on the evolution of the internal network structure ofthe three major urban agglomerations in China[D].Fujian Normal University, 2021. https://doi.org/10.27019/d.cnki.gfjsu.2021.000573

Response 4: This section was revised by ourselves, and then we consulted experts and scholars in related fields for guidance on the content of the article; we do not list the changes here, but in the revised document has been marked out.

---

## [Decision Letter · Decision Letter 1]

12 Dec 2022

Research on the characteristics and influencing factors of the Beijing-Tianjin-Hebei urban network structure from the perspective of listed manufacturing enterprises

PONE-D-22-26849R1

Dear Dr. Miao,

We’re pleased to inform you that your manuscript has been judged scientifically suitable for publication and will be formally accepted for publication once it meets all outstanding technical requirements.

Kind regards,

Jun Yang

Academic Editor

PLOS ONE

Additional Editor Comments (optional):

Accept

Reviewers' comments:

Reviewer's Responses to Questions

**Comments to the Author**

1. If the authors have adequately addressed your comments raised in a previous round of review and you feel that this manuscript is now acceptable for publication, you may indicate that here to bypass the “Comments to the Author” section, enter your conflict of interest statement in the “Confidential to Editor” section, and submit your "Accept" recommendation.

Reviewer #1: All comments have been addressed

Reviewer #2: All comments have been addressed

2. Is the manuscript technically sound, and do the data support the conclusions?

Reviewer #1: Yes

Reviewer #2: Yes

3. Has the statistical analysis been performed appropriately and rigorously? 

Reviewer #1: Yes

Reviewer #2: N/A

4. Have the authors made all data underlying the findings in their manuscript fully available?

Reviewer #1: Yes

Reviewer #2: Yes

5. Is the manuscript presented in an intelligible fashion and written in standard English?

Reviewer #1: Yes

Reviewer #2: Yes

6. Review Comments to the Author

Reviewer #1: The authors have adequately addressed most of the comments raised in a previous round of my review and I think that this manuscript should be accepted for publication.

Reviewer #2: The author have conducted a fully and effective repsonse to my comments and suggestions. Consequently, it is suggested to be published.

7. PLOS authors have the option to publish the peer review history of their article (what does this mean?). If published, this will include your full peer review and any attached files.

Reviewer #1: No

Reviewer #2: No

---

## [Editor Report · Acceptance letter]

19 Dec 2022

PONE-D-22-26849R1 

Research on the characteristics and influencing factors of the Beijing-Tianjin-Hebei urban network structure from the perspective of listed manufacturing enterprises 

Dear Dr. Miao:

I'm pleased to inform you that your manuscript has been deemed suitable for publication in PLOS ONE. Congratulations! Your manuscript is now with our production department. 

Kind regards, 

on behalf of

Dr. Jun Yang 

Academic Editor

PLOS ONE